# Dietary Pistachio (*Pistacia vera* L.) Beneficially Alters Fatty Acid Profiles in Streptozotocin-Induced Diabetic Rat

**Ioanna Prapa** [1], **Amalia E. Yanni** [2,*], **Anastasios Nikolaou** [1], **Nikolaos Kostomitsopoulos** [3], **Nick Kalogeropoulos** [2], **Eugenia Bezirtzoglou** [4], **Vaios T. Karathanos** [2] and **Yiannis Kourkoutas** [1,*]

1   Laboratory of Applied Microbiology and Biotechnology, Department of Molecular Biology and Genetics, Democritus University of Thrace, Dragana, 68100 Alexandroupolis, Greece; ioannaprap@gmail.com (I.P.); anikol@mbg.duth.gr (A.N.)

2   Laboratory of Chemistry, Biochemistry, Physical Chemistry of Foods, Department of Nutrition and Dietetics, Harokopio University of Athens, 17671 Athens, Greece; nickal@hua.gr (N.K.); vkarath@hua.gr (V.T.K.)

3   Laboratory Animal Facility, Biomedical Research Foundation of the Academy of Athens, 11527 Athens, Greece; nkostom@bioacademy.gr

4   Laboratory of Hygiene and Environmental Protection, Department of Medicine, Democritus University of Thrace, Dragana, 68100 Alexandroupolis, Greece; empezirt@yahoo.gr

*   Correspondence: ayanni@hua.gr (A.E.Y.); ikourkou@mbg.duth.gr (Y.K.); Tel.: +30-2109-549-174 (A.E.Y.); +30-2551-030-633 (Y.K.)

**Featured Application: The present study could contribute to the design of dietary patterns with beneficial effects on fatty acids profile and gut microbiota of Type-1 diabetic patients.**

**Abstract:** Type 1 Diabetes (T1D) onset has been associated with diet, among other environmental factors. Adipose tissue and the gut have an impact on β-cell biology, influencing their function. Dietary ingredients affect fatty acid profiles of visceral adipose tissue (VAT) and plasma, as well as SCFAs production after microbial fermentation. Pistachios are a rich source of oleic acid, known for their anti-inflammatory actions and favorably affect gut microbiota composition. The purpose of the study was to examine plasma and VAT fatty acids profiles as well as fecal SCFAs after dietary intervention with pistachio nuts in streptozotocin-induced diabetic rats. Plasma and VAT fatty acids were determined by GC-MS and SCFAs by HPLC. After 4 weeks of pistachio consumption, MUFA and especially oleic acid were increased in plasma and VAT of diabetic rats while PUFA, total ω6 and especially 18:2ω6, were decreased. Lactic acid, the major end-product of beneficial intestinal microorganisms, such as lactobacilli, was elevated in healthy groups, while decreased levels of isovaleric acid were recorded in healthy and diabetic groups following the pistachio diet. Our results reveal possible beneficial relationships between pistachio nut consumption, lipid profiles and intestinal health in the disease state of T1D.

**Keywords:** type 1 diabetes; streptozotocin-induced diabetes; pistachio nuts; fatty acids; inflammation; short-chain fatty acids

## 1. Introduction

Diabetes mellitus (DM) represents a major health concern since its prevalence is continuously increasing worldwide. In 1980, the incidence of DM was 108 million, while in 2019, it was raised to approximately 463 million cases (adults older than 20 years). It is estimated that by 2045 the number of people suffering from the disease will reach 700 million, according to the World Health Organization [1] and the International Diabetes Federation [2].

Type 1 diabetes (T1D), or insulin-dependent DM, is a disease attacking mainly young people (children and adolescents) and constitutes about 5–10% of all cases of diabetes [1,3]. It is evoked by the autoimmune destruction of pancreatic β-cells, resulting in integral

insulin deficiency and disturbed glucose metabolism [4]. Besides genetic predisposition, including particular haplotypes of the human leucocyte antigen [5], other factors, such as infections, caesarean section instead of normal childbirth, dietary habits and the overuse of antibiotics have been associated with T1D [6]. However, the exact mechanisms through which these parameters contribute to the onset and the progression of the disease are not well documented [7]. Diet is a modifiable factor of pivotal importance that can significantly contribute to accomplishing better management of the disease [8].

Pistachios (*Pistacia vera* L.) are a mainstay of the Mediterranean diet due to their excellent sensory characteristics, as well as their high nutritional quality, attributed to their content in dietary fiber, vitamins, minerals and phytochemicals, known for their health-promoting effects [9]. Pistachios contain high amounts (73%) of monounsaturated fatty acids (MUFA), especially oleic acid (18:1ω9) (70%), known for its cardioprotective, hypocholesterolemic, hypoglycemic and anti-inflammatory properties [10,11], as well as polyunsaturated fatty acids (PUFA) (16%) [12]. The presence of unsaturated fatty acids, which is accompanied by low saturated fatty acids (SFA) content (11%), renders pistachio an optimal choice for maintaining a healthy lipid profile [13,14]. Several studies imply that SFA promotes insulin resistance and diabetes development, whilst MUFA and especially ω3 PUFA can retreat these harmful effects [15,16].

Dietary patterns influence the composition of gut microbiota, which is directly linked with host homeostasis [17]. Mediterranean diet-induced changes in intestinal microbiota give rise to greater production of short-chain fatty acids (SCFAs), particularly of butyrate [18]. The concentration of SCFAs in the gastrointestinal (GI) tract and peripheral circulation may predispose to or prevent pathological conditions, such as diabetes and other metabolic diseases. Furthermore, these acids exert pivotal functions on the anti-inflammatory responses in the intestinal region and participate in multiple signalling pathways of lipid and glucose metabolism [19]. A previous study [20] has demonstrated beneficial alterations in the gut microbiome of T1D rats after pistachio consumption, while favorable effects have also been supported by other researchers in obese mice [21], revealing the presence of underlying mechanisms involved in anti-inflammatory actions [22].

Adipose tissue is a complex endocrine organ that regulates metabolic homeostasis. Adipocytes synthesize and express several mediators, among them inflammatory cytokines, which have been suggested to be associated with the onset of T1D and the progression of its implications [23]. Recently, VAT inflammation and dysfunction were proposed to be linked with the progression of the disease of T1D [24]. From this point of view, VAT is considered a target organ and the identification of mechanisms linking metabolic pathways with β-cell autoimmunity is crucial for the development of new therapeutic approaches for the management of the disease.

Fatty acid profiles in VAT and plasma partly reflect the composition of dietary fat but also the activities of enzymes responsible for the synthesis, desaturation and elongation of fatty acids [25]. Dietary ingredients determine MUFA and PUFA content and shape the gut microbiota and, therefore, the SCFAs concentration. Of note, dietary-induced alterations of the gut microbiota convert the latter into a critical determinant, the modulation of which can enhance or nullify T1D progression [26]. Both gut and adipose tissue affect β-pancreatic cell homeostasis and function in healthy and diabetic states [27]. Mapping the differences observed in fatty acid profiles at various sites after dietary interventions with a promising food or ingredient compromises the first step in elucidating the potential targets that could ameliorate the metabolic profile and benefit T1D patients [28].

The streptozotocin (STZ)-induced diabetic rat is an animal model that has been extensively employed for the study of diabetes and its complications due to the fact that the pathology mimics that of human T1D and is characterized by chronic pancreatic cell inflammation, insulitis and insulin deficiency [29]. Furthermore, the STZ-induced animal model is more easily handled and has lower cost compared to genetically modified models [29].

Thus, the purpose of the present study was to investigate whether a dietary intervention with pistachio nuts causes favorable alterations in the fatty acid profiles of plasma, VAT and fecal SCFAs of healthy and STZ-induced (T1D) rats.

## 2. Materials and Methods

### 2.1. In Vivo Study Design

Twenty-four male RccHan®:WIST rats (350–400 g bw) were randomized into four groups (six animals per group) according to their dietary regimen, i.e., healthy animals that received the control (CD) or pistachio diet (PD) and STZ-diabetic animals that received the control (DCD) or pistachio diet (DPD) for 4 weeks [20]. Diabetes was induced after intraperitoneal injection of 40 mg/kg bw STZ in 0.1 M citrate buffer pH 4.5. The animals were considered diabetic when postprandial glucose was >250 mg/dL. The data for hyperglycemia and hypoinsulinemia for T1D are reported in a previous study [20]. The amount of pistachio was added to a total yield of 100 g/kg fat in the pistachio diet. The percentage of pistachios in the food was 8% *w/w*, and the daily amount of food provided to the animals was 20 g (standard food and pistachios). The amount of pistachios provided to the animals was based on the design of relevant studies, including dietary interventions in animal models [30] and humans, so that the energy content would be suitable for the human diet [31,32]. The control diet was supplemented with corn oil in order to equalize the amount of fat and caloric content of the pistachio diet. Corn oil was used in the control diet because it has low oleic acid content, which is the predominant fatty acid in pistachio nuts.

Fresh fecal samples were collected at 0 (baseline) and 4th week (end) and stored at −80 °C until SCFAs analysis. At the end of the experimental period, the animals were sacrificed by an overdose of isoflurane. Blood samples were collected by cardiac puncture, and EDTA plasma was separated by centrifugation at 4 °C at $1000 \times g$ for 10 min. VAT was isolated and placed immediately on dry ice. Plasma and VAT were stored at −80 °C until the analysis of fatty acids.

Animal experimentation was reviewed and approved by the Veterinary Directorate of the Athens Prefecture (Ref. Number 2057/05-04-2017), the Committee on Research Ethics of Democritus University of Thrace (Ref. Number 9254/386/22-05-2020), and conducted in compliance with the European Directive 2010/63. All experimentations took place in the Centre of Experimental Surgery of the Biomedical Research Foundation of the Academy of Athens (BRFAA). The animals were housed in accordance with the European legal framework and the international guidelines existing for the protection of animals used for scientific purposes.

### 2.2. Analysis of Plasma Fatty Acids

Plasma fatty acids were determined by the method of Lepage and Roy [33] with slight modifications. Briefly, 200 μL of plasma samples were lyophilized (Scientz-18N, Ningbo Scientz Biotechnology Co., Ltd., Ningbo, China.) for 4 h. After lyophilization, 3 mL of 4:1 $CH_3OH/C_7H_8$ solution were added, followed by the step-by-step addition of 200 μL $CH_3COCl$ in three doses and subsequent vortex mixing. The samples were subjected to methanolysis in a water bath (Memmert WB14, Schwabach, Germany) at 100 °C for 30 min and then cooled at room temperature. A total of 5 mL of $K_2CO_3$ 6% *w/v* solution was slowly added in order to terminate the reaction and neutralize the mixture. The samples were centrifuged (Hettich Universal 32, Tuttlingen, Germany) at $1000 \times g$ for 10 min and the supernatant organic phase of toluene, including fatty acids methyl esters (FAME), was transferred to GC vials, which were sealed and stored at −80 °C until analysis.

FAMEs were separated on a 60 m × 0.25 mm capillary column (0.25 μm film thickness) DB-23 J&W (Agilent Technologies, Santa Clara, CA, USA), using a Hewlett-Packard 6890 Gas Chromatograph (Agilent Technologies, Waldbronn, Germany) equipped with an MSD-5972 mass selective detector. The analytical conditions were as follows: volume injected; 1 μL, split ratio; 1:20, carrier gas; high-purity helium (0.8 mL/min), injector temperature;

230 °C, MSD transfer line; 280 °C, oven temperature; from 130 to 250 °C with stepped temperature program, total run time; 60 min.

Identification of the FAME peaks was accomplished using a mixed FAME standard (Sigma L9405, St Louis, MO, USA) and by reference to the NIST mass spectra library. The mixed FAME standard was injected periodically to check for any changes in retention times, while it also served for the calculation of response factors, which were applied to the areas derived from the chromatographic traces.

### 2.3. Analysis of Adipose Tissue Fatty Acids

Total fatty acids were extracted as previously described [34]. In brief, 100 mg of tissue samples were thawed, transferred to 10 mL screw-capped tubes and saponified with 0.5 M NaOH methanolic solution. The FAME was prepared with 14% boron trifluoride in methanol and was recovered with hexane containing 100 ppm BHT after washing the organic layer with saturated NaCl solution. The hexane layer was transferred to GC vials, which were sealed and stored at −80 °C until analysis.

GC-MS analysis was carried out as described above for the analysis of plasma fatty acids.

### 2.4. Analysis of Fecal SCFAs and Lactic Acid

Fatty acid purification and extraction from feces was performed as described by Chaia and Zárate [35], with slight modifications. Acidification and homogenization (Snijders Vortex 34524, Snijders Scientific, Tilburg, Holland) of feces (0.20–0.25 g) in 1 mL $H_2SO_4$ (0.15 mM) was performed, followed by centrifugation (9000× $g$, 4 °C, 20 min) (K241R Medium Prime Centrifuge, Centurion Scientific, Chichester, UK). The supernatant was treated with trichloroacetic acid (85%) at a concentration of 75 µL/2 mL and incubated on wet ice for 45 min. Protein precipitation was completed after centrifugation (17,000× $g$, 20 °C, 20 min), and the supernatant was filtered twice in a nylon filter 0.22 µm. The samples were stored at −20 °C until analysis.

SCFAs (acetic, propionic, butyric, isobutyric, valeric and isovaleric acids), as well as lactic acid concentrations, were determined by HPLC, using a Shimadzu chromatography system (Shimadzu Corp., Duisburg, Germany) equipped with a Nucleogel ION 300 OA column (Macherey-Nagel, Düren, Germany), a DGU- 20A5R degassing unit, a LC-20AD pump, a CTO-20AC oven at 85 °C and a RID-10A refractive index detector, as previously described [36], with some modifications. In brief, a solution of 0.049 g/L $H_2SO_4$ was used as the mobile phase at 0.4 mL/min. A total of 20 µL of each sample were injected directly into the column, and the detector cell temperature was set at 60 °C. SCFAs and lactic acid concentrations were calculated using standard curves prepared by standard solutions ($R^2 \geq 0.99$).

Fecal SCFAs and lactate concentrations were expressed as mean µmol per gram of feces, using the following equation, as described by Huda-Faujan et al. [37]: SCFAs/lactate (µmol/g) = [organic acid in fecal contents (mmol/mL) * $V_d$ (mL) * 1000]/weight of feces (g), where: $V_d$ = Total Volume of Dilution.

### 2.5. Statistical Analysis

The data are expressed as mean ± SEM. Statistical analysis was performed using SPSS 21.0 statistical software. Variables were tested for normal distribution with the Kolmogorov–Smirnov test. Factorial analysis of variance (ANOVA) coupled with the Bonferroni post-hoc test was used to compare VAT and plasma individual fatty acid content in the four groups of animals and amounts of fecal SCFAs samples. The level of statistical significance was set at $p < 0.05$.

## 3. Results

### 3.1. Plasma Fatty Acid Profiles

The highest percentages of fatty acids identified in the plasma samples of all groups were 18:2ω6, 20:4ω6, 16:0, 18:0 and 18:1ω9 (Table 1). Fatty acids 16:1 and 17:0 were not

detected in the two groups of diabetic animals, whereas 18:3ω6 was not detected in the groups of animals supplemented with pistachio. Total PUFA and SFA were more abundant ($p < 0.05$) compared to MUFA in all groups.

**Table 1.** Plasma fatty acid profile in the four groups of animals at the end of the four-week dietary intervention.

| Fatty Acids (%) | CD | PD | DCD | DPD |
|---|---|---|---|---|
| 16:0 | 17.26 ± 0.36 | 17.64 ± 0.51 | 16.92 ± 1.84 | 18.04 ± 0.53 |
| 16:1 | 0.30 ± 0.02 | 0.35 ± 0.05 | N/D | N/D |
| 17:0 | 0.30 ± 0.01 | 0.31 ± 0.03 | N/D | N/D |
| 18:0 | 11.27 ± 0.65 | 11.94 ± 0.55 | 11.32 ± 1.29 | 13.83 ± 0.33 |
| 18:1ω9 | 10.72 ± 0.61 | 10.90 ± 1.37 | 9.77 ± 1.43 | 14.43 [a] ± 0.80 |
| 18:2ω6 | 28.46 ± 0.95 | 21.50 ± 0.55 | 26.71 ± 3.78 | 22.36 ± 0.95 |
| 18:3ω6 | 1.06 ± 0.06 | N/D | 1.20 ± 0.13 | N/D |
| 20:4ω6 | 23.26 ± 1.14 | 24.08 ± 1.77 | 18.47 ± 3.19 | 23.33 ± 2.06 |
| 22:6ω3 | 1.66 ± 0.06 | 1.87 ± 0.12 | 1.81 ± 0.28 | 1.99 ± 0.18 |
| SFA | 28.83 ± 0.80 | 29.90 ± 0.96 | 28.24 ± 2.96 | 31.87 ± 0.85 |
| MUFA | 11.02 ± 0.63 | 11.25 ± 1.33 | 9.77 ± 1.43 | 14.43 [a] ± 0.80 |
| PUFA | 54.43 ± 1.16 | 47.44 ± 1.72 | 48.19 ± 4.28 | 47.68 ± 1.38 |
| ω6 | 52.77 ± 1.11 | 45.57 ± 1.67 | 46.38 ± 4.06 | 45.69 ± 1.47 |
| ω3 | 1.66 ± 0.06 | 1.87 ± 0.12 | 1.81 ± 0.28 | 1.99 ± 0.18 |
| ω6/ω3 | 31.93 ± 0.70 | 24.37 ± 1.32 | 28.66 ± 3.83 | 24.01 ± 2.38 |

Values are expressed as mean ± SEM, [a] $p < 0.05$ vs. DCD. N/D: Non-Detected, CD: Control Diet, PD: Pistachio Diet, DCD: Diabetic rats Control Diet, DPD: Diabetic rats Pistachio Diet, SFA: saturated fatty acids, MUFA: monounsaturated fatty acids, PUFA: polyunsaturated fatty acids.

In diabetic rats that received the pistachio diet, MUFA was significantly higher ($p < 0.05$) compared to diabetic animals that received the diet supplemented with corn oil. SFA and PUFA were at similar levels in all animal groups ($p > 0.05$).

### 3.2. Adipose Tissue Fatty Acid Profiles

Among all fatty acids detected in adipose tissue samples, 18:2ω6, 18:1ω9 and 16:0 were the most abundant ($p < 0.05$) in all groups (Table 2). Dietary supplementation with pistachio, along with the T1D state, significantly affected the relative composition of adipose tissue fatty acids (Table 2). A typical chromatograph of FAME from a VAT sample of a diabetic rat receiving the pistachio diet is presented in Figure 1.

**Table 2.** Fatty acid profile of visceral adipose tissue in the four groups of animals.

| Fatty Acids (%) | CD | PD | DCD | DPD |
|---|---|---|---|---|
| 12:0 | 0.03 ± 0.04 | 0.04 ± 0.01 | 0.02 ± 0.00 | 0.03 ± 0.00 |
| 14:0 | 0.39 ± 0.01 | 0.40 ± 0.03 | 0.34 ± 0.03 | 0.35 ± 0.02 |
| 15:0 | 0.24 ± 0.01 | 0.23 ± 0.01 | 0.20 ± 0.01 | 0.20 ± 0.01 |
| 15:1 | 0.05 ± 0.00 | 0.01 ± 0.00 | 0.002 ± 0.00 | 0.05 ± 0.01 |
| 16:0 | 17.91 ± 0.24 | 17.61 ± 0.31 | 15.16 [b,c] ± 0.78 | 14.92 [b,c] ± 0.56 |
| 16:1 | 1.26 ± 0.05 | 1.42 ± 0.20 | 0.90 ± 0.29 | 0.70 ± 0.10 |
| 17:0 | 0.22 ± 0.01 | 0.21 ± 0.01 | 0.23 ± 0.01 | 0.22 ± 0.01 |
| 17:1 | 0.09 ± 0.01 | 0.08 ± 0.01 | 0.06 ± 0.01 | 0.06 ± 0.01 |
| 18:0 | 2.67 ± 0.04 | 2.56 ± 0.07 | 3.06 ± 0.18 | 3.07 ± 0.16 |
| 18:1ω9 | 25.53 ± 0.10 | 30.22 [b] ± 0.80 | 26.80 [c] ± 0.31 | 31.59 [a,b] ± 0.67 |
| 18:1ω7 | 2.87 ± 0.05 | 3.01 ± 0.07 | 3.11 ± 0.20 | 3.22 ± 0.08 |
| 18:2ω6 | 43.35 ± 0.31 | 39.33 [b] ± 0.25 | 44.80 [c] ± 0.70 | 40.81 [a,b] ± 0.55 |
| 18:3ω6 | 0.11 ± 0.01 | 0.12 ± 0.01 | 0.10 ± 0.01 | 0.11 ± 0.03 |

**Table 2.** *Cont.*

| Fatty Acids (%) | CD | PD | DCD | DPD |
|---|---|---|---|---|
| 18:3ω3 (ALA) | 1.92 ± 0.04 | 1.74 ± 0.09 | 1.26 [b] ± 0.18 | 1.07 [b,c] ± 0.08 |
| 20:0 | 0.10 ± 0.01 | 0.10 ± 0.00 | 0.16 ± 0.03 | 0.16 ± 0.01 |
| 20:1ω9 | 0.23 ± 0.00 | 0.24 ± 0.00 | 0.34 ± 0.05 | 0.34 ± 0.03 |
| 20:2ω6 | 0.32 ± 0.01 | 0.30 ± 0.03 | 0.38 ± 0.04 | 0.35 ± 0.05 |
| 20:3ω6 | 0.17 ± 0.02 | 0.15 ± 0.01 | 0.23± 0.02 | 0.21± 0.03 |
| 20:4ω6 | 0.73 ± 0.05 | 0.64 ± 0.08 | 0.59 ± 0.08 | 0.50 ±0.13 |
| 24:0 | 0.12 ± 0.01 | 0.12 ± 0.01 | 0.20 ± 0.04 | 0.19 ±0.05 |
| 22:6ω3 (DHA) | 0.11 ± 0.01 | 0.11 ± 0.03 | 0.15± 0.04 | 0.15 ± 0.06 |
| SFA | 21.68 ± 0.24 | 21.26 ± 0.35 | 19.37 [b,c] ± 0.62 | 19.14 [b,c] ± 0.38 |
| MUFA | 30.05 ± 0.13 | 35.03 [b] ± 0.58 | 31.26 [c] ± 0.47 | 35.95 [a,b] ± 0.61 |
| PUFA | 46.73 ± 0.21 | 42.39 [b] ± 0.30 | 47.51 [c] ± 0.62 | 43.20 [a,b] ± 0.71 |
| ω6 | 44.69 ± 0.24 | 40.54 [b] ± 0.24 | 46.10 [c] ± 0.64 | 41.98 [a,b] ± 0.68 |
| ω3 | 2.04 ± 0.05 | 1.85 ± 0.12 | 1.40 [b] ± 0.10 | 1.22 [b,c] ± 0.09 |
| ω6/ω3 | 22.01 ± 0.60 | 22.32 ± 1.20 | 36.20 [b,c] ± 5.49 | 35.23 ± 2.71 |

Values are expressed as mean ± SEM, [a] $p < 0.05$ vs. DCD, [b] $p < 0.05$ vs. CD, [c] $p < 0.05$ vs. PD. CD: Control Diet, PD: Pistachio Diet, DCD: Diabetic rats Control Diet, DPD: Diabetic rats Pistachio Diet, ALA: alpha-linolenic acid, DHA: Docosahexaenoic acid, SFA: saturated fatty acids, MUFA: monounsaturated fatty acids, PUFA: polyunsaturated fatty acids.

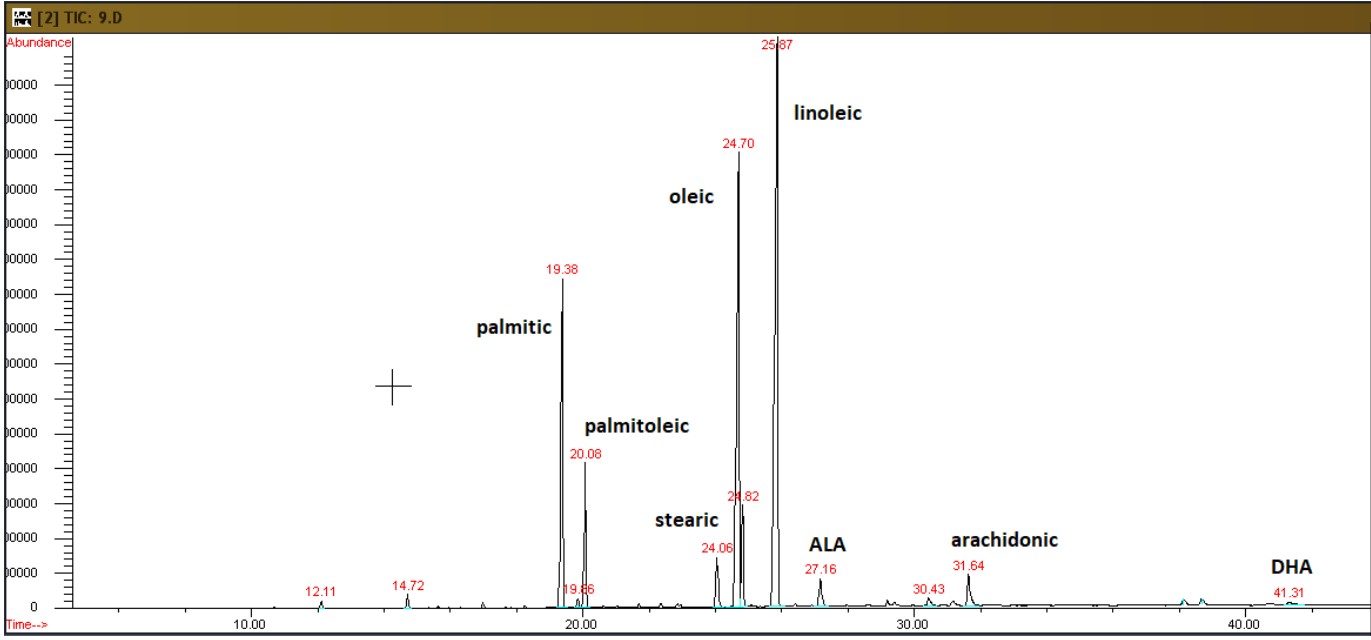

**Figure 1.** A typical chromatogram of fatty acids methyl esters from a VAT sample of a diabetic rat receiving the pistachio diet.

Diabetic groups exerted decreased ($p < 0.05$) concentrations of palmitic (16:0) and a-linolenic acid (18:3ω3 or ALA) (DCD compared to CD group and DPD compared to PD). Oleic acid (18:1ω9) levels were higher ($p < 0.05$), whereas linoleic acid (18:2ω6) were lower ($p < 0.05$) in PD and DPD groups compared to CD and DCD, respectively.

The total SFA of DCD and DPD groups were lower ($p < 0.05$) compared to CD and PD groups. After the pistachio diet, in groups PD and DPD, MUFA was increased and PUFA was decreased compared to CD and DCD ($p < 0.05$). Total ω6 polyunsaturated fatty acids were lower ($p < 0.05$) in PD and DPD compared to CD and DCD groups. An elevated ratio of ω6/ω3 was recorded in DCD compared to the CD group ($p < 0.05$), as well as in DPD compared to PD but without reaching statistical significance ($p > 0.05$).

### 3.3. Fecal SCFAs and Lactic Acid

Dietary supplementation with pistachio and the state of diabetes led to alterations in the levels of SCFAs and lactate in feces, as shown in Table 3. Acetic acid had the higher concentration of all fecal SCFAs detected in all cases ($p < 0.05$). In diabetic rats, lactic acid was elevated at week 0 (baseline) compared to healthy ones ($p < 0.05$ in DCD compared to CD and DPD compared to PD). In both diabetic groups, the levels of lactate were decreased ($p < 0.05$) after 4 weeks compared to baseline. At the end of the study, dietary pistachio led to higher ($p < 0.05$) levels of lactic acid in the PD group compared to baseline, whereas such a difference was not observed in the CD group ($p > 0.05$).

**Table 3.** Fecal short-chain fatty acids (SCFAs) in the four groups of animals, at baseline and at the end of the dietary intervention.

| | CD | | PD | | DCD | | DPD | |
|---|---|---|---|---|---|---|---|---|
| | **Baseline** | **End** | **Baseline** | **End** | **Baseline** | **End** | **Baseline** | **End** |
| lactic acid | 0.798 ± 0.063 | 0.921 ± 0.216 | 0.760 ± 0.056 | 1.345 [c] ± 0.625 | 1.429 ± 0.314 | 0.819 [c] ± 0.109 | 1.508 ± 0.490 | 0.711 [c] ± 0.073 |
| acetic acid | 5.224 ± 0.785 | 7.274 ± 2.169 | 5.303 ± 0.671 | 6.617 ± 0.720 | 5.744 ± 2.963 | 9.171 ± 2.410 | 6.424 ± 2.764 | 5.845 ± 1.827 |
| propionic acid | 0.911 ± 0.150 | 0.960 ± 0.145 | 0.965 ± 0.073 | 1.078 ± 0.241 | 0.916 ± 0.438 | 1.225 ± 0.284 | 1.019 ± 0.407 | 0.766 ± 0.124 |
| isobutyric acid | 0.199 ± 0.012 | 0.213 ± 0.038 | 0.206 ± 0.012 | 0.249 ± 0.046 | 0.173 ± 0.026 | 0.215 ± 0.039 | 0.164 ± 0.020 | 0.180 ± 0.024 |
| butyric acid | 0.937 ± 0.149 | 0.845 ± 0.159 | 0.834 ± 0.175 | 0.936 ± 0.312 | 0.606 ± 0.196 | 0.764 ± 0.213 | 0.653 ± 0.223 | 0.580 ± 0.198 |
| isovaleric acid | 0.040 ± 0.009 | 0.074 ± 0.016 | 0.056 ± 0.010 | 0.034 [b] ± 0.012 | 0.030 ± 0.016 | 0.068 ± 0.048 | 0.033 ± 0.013 | 0.026 [a] ±0.014 |
| valeric acid | 0.109 ± 0.033 | 0.191 ± 0.031 | 0.135 ± 0.034 | 0.134 ± 0.039 | 0.120 ± 0.041 | 0.191 ± 0.067 | 0.183 ± 0.054 | 0.182 ± 0.064 |

Values are expressed as mean ± SEM, [a] $p < 0.05$ vs. DCD, [b] $p < 0.05$ vs. CD, [c] $p < 0.05$ compared to baseline values of same group. CD: Control Diet, PD: Pistachio Diet, DCD: Diabetic rats Control Diet, DPD: Diabetic rats Pistachio Diet.

Acetic, propionic, butyric and isobutyric acids were at similar concentrations in all groups ($p > 0.05$). In the fourth week of the study, isovaleric acid levels were detected in higher amounts in the CD and DCD groups compared to PD ($p < 0.05$) and DPD ($p < 0.05$), respectively. A representative HPLC analysis chromatograph of lactic acid and SCFAs determination is shown in Figure 2.

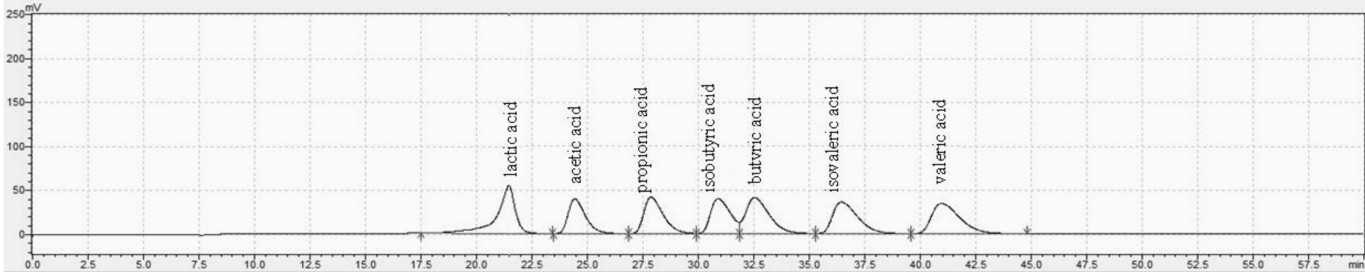

**Figure 2.** A representative HPLC chromatogram of lactic acid and SCFAs determination in rat feces, using a Refractive Index Detector (RID-10A).

### 4. Discussion

T1D is a chronic autoimmune disease that is strongly associated with inflammation. Innate immunity, as well as inflammatory mediators, are implicated in T1D more notably than initially assumed [23]. VAT inflammation and dysfunction have recently been recognized as early mechanisms triggering β-cell autoimmunity in T1D [24,38]. These pro-inflammatory actions are associated with macrophage recruitment, upregulation of pro-inflammatory cytokines and oxidative stress responses, apoptosis and downregulation of adipokines and the release of free fatty acids and molecules that affect insulin signaling. Insulin acts on adipose tissue by enhancing glucose absorption, triglyceride synthesis, suppressing triglyceride hydrolysis and the release of free fatty acids and glycerol into the bloodstream [39]. Adipocytes express key receptors for SCFAs, and adipose homeostasis can be modulated by SCFA profile and the composition of gut microbiota. Butyrate has been shown to inhibit NFκB activation and moderate VAT inflammation [40]. In addition,

gut and adipose tissue have an impact on β-cell biology, influencing the function of β-cells under normal and diabetogenic conditions [27]. Taking into consideration these inter-organ communications, the identification of the fatty acid profile in VAT and plasma as well as SCFAs in feces could provide important information regarding the disease of T1D and the possible beneficial effects of dietary interventions.

Dietary ingredients affect glycemic control, among others, and thus can contribute to the progress of diabetes complications [28]. In patients with severe T1D, increased plasma lipids are detected, which can be rendered to normal after adequate insulin therapy [41]. The management of dietary fat and especially the decrease of total fat, saturated fat and trans-fatty acids intake is proposed [42]. MUFA and PUFA can be used as alternatives to maintain lipid intake within accepted ranges or to ameliorate the lipid profile [41]. Therefore, nutrition therapy is crucial for T1D management and inflammation.

In the present study, the concentration of plasma MUFA was increased in diabetic rats following the pistachio diet compared to those fed with the standard diet, demonstrating that dietary patterns affected the lipid profile in a favorable manner, as elevated plasma MUFA following a high-MUFA dietary intervention has been associated with improvements in insulin sensitivity [43]. What is more, in DPD rats, dietary pistachio enhanced the concentration of oleic acid ($18:1\omega9$) compared to rats fed a standard diet. Oleic acid is found in olive oil-rich diets, such as the Mediterranean diet, and it is suggested that it can improve blood sugar control and insulin sensitivity [44,45]. The beneficial effects of diets high in MUFA are well documented [46], while the composition of the lipid profile in plasma and VAT is at least partially reflected by the diet.

It is widely accepted that dietary fat and Its fatty acid composition interfere with lipid metabolism in VAT [47,48]. Triacylglycerols of VAT mainly include $18:1\omega9$, 16:0 and $18:2\omega6$ [49], which were also confirmed in our study, where in all groups, the above-mentioned fatty acids were detected.

In the VAT of diabetic rats of both groups, decreased concentrations of 16:0, SFA and $18:3\omega3$ compared to healthy animals were detected, which is in accordance with the results reported by other studies [50,51]. Pistachio consumption had no effect on these alterations and, therefore, these can be attributed to the diabetic state. The decrease in total SFA may imply that the diabetic state worsens their biosynthesis and elongation process, but it may also be due to a decrease in palmitic acid, which cannot be further elongated to other saturated fatty acids at the same rate [52]. The predominance of 16:0 fatty acid, which is lower in diabetic animals compared to the animals of the control group, leads to a significant reduction in total SFA seen in diabetic groups, while a non-significant decreasing trend prevails in most of the other SFA.

Dietary pistachio led to significantly increased levels of $18:1\omega9$ and MUFA, whereas $18:2\omega6$, total ω6 and PUFA were decreased in the VAT of diabetic and healthy animals compared to groups fed with the standard diet. These alterations seem to be an outcome of pistachio supplementation, as they occur in both PD and DPD groups, regardless of the disease state. Pistachio is rich in oleic acid ($18:1\omega9$) (and particularly 69.6% in total FAs) and MUFA (73%) [13]; therefore, the dietary intake of these fatty acids leads to their elevated presence in the VAT of PD and DPD, compared to CD and DCD groups. Other MUFA, apart from $18:1\omega9$, were detected at similar levels in all groups. Increased MUFA in VAT has been reported after a 4-week olive-oil diet in rats [53], accompanied by an increase in oleic acid, results that underline that pistachio supplementation could possibly affect fatty acid profile in a similar manner, which has been associated with an ameliorated glycemic response in T1D state [54].

The control groups, CD and DCD, had significantly elevated amounts of $18:2\omega6$ compared to groups that received the pistachio nuts (PD and DPD). Although pistachio contains adequate amounts of PUFA and particularly $18:2\omega6$, their content, as well as that of total ω6, was not enhanced in the VAT of PD and DPD groups. This is probably due to the supplementation of the control diet with corn oil (in order to equilibrate total fat and energy content of the two different diets), which is also rich in $18:2\omega6$.

Levels of total ω3 FAs (i.e., 22:6ω3 and 18:3ω3) were significantly decreased in diabetic compared to healthy animals. More specifically, 22:6ω3 was found at similar levels in all groups, but 18:3ω3 concentrations were lower in DCD and DPD compared to healthy ones, indicating that 18:3ω3 levels determine the difference in the total ω3. An elevated ratio of ω6/ω3, which has been associated with insulin resistance [55,56], was found in the DCD group compared to CD, as well as in the DPD group compared to PD, with the latter exerting a slightly better trend. VAT performs constant metabolic actions as it is implicated in the release of fatty acids into circulation [40]; therefore, its composition of fatty acids contributes to conditions such as systemic inflammation and dyslipidemia [57], rendering VAT a key organ in the disorder of T1D [23,24].

It has been widely reported that dietary interventions are able to modulate gut microbiota and SCFA concentrations, eventually attenuating low-grade inflammation and improving glucose metabolism [58]. In this manner, fecal SCFAs and lactic acid levels were determined. Acetic acid was the predominant SCFA detected in all groups, in accordance with other studies [59]. In T1D mice, the supplementation with diets that resulted in the release of large amounts of acetate or butyrate after bacterial fermentation in the large intestine reduced plasma concentration of diabetogenic molecules, and a general attenuation of autoimmune response was recorded [60].

Lactic acid is produced by the host cells in inflammatory diseases [61], but it can also be produced by intestinal microbiota [62]. Interestingly, the fecal lactic acid concentration was significantly higher in diabetic groups at baseline compared to healthy groups. However, in the fourth week of the study, the lactic acid levels were lower in both diabetic groups and rendered to normal levels. This could probably be attributed to the injection of STZ and the induction of diabetes at the beginning of the study. On the other hand, in healthy rats, after 4 weeks of pistachio supplementation, the lactic acid concentration was increased compared to baseline values, whereas such difference was not observed in groups fed with the standard diet. Lactic acid is the major metabolic product of lactic acid bacteria, the beneficial organisms that are hosted in the GI tract [63], and as an intermediate organic acid in the bacterial fermentation of carbohydrates, it is further converted to SCFAs. It has been shown that after a prebiotic diet for 60 days, fecal lactic acid concentration was increased [64]. As previously reported [20], pistachio led to an increased presence of lactobacilli, a genus that produces lactic acid in the colon through metabolic fermentation of glucose [63]. However, lactic acid content may rise as a result of inflammation and has been suggested to cause inflammation [65]. It can be accumulated under conditions of dysbiosis, for example, in severe colitis [65], inflammatory bowel disease [37,66] and appendicitis [67], and this can be due to the curtailment of the growth of lactate-utilizing bacteria at reduced pH conditions [68]. Further studies are needed in order to shed light on the role of lactic acid in STZ-induced diabetes.

Pistachio also led to decreased levels of isovaleric acid in both healthy and diabetic rats compared to control diet-supplemented rats. In a double-blind, cross-over randomized controlled trial, fecal isovalerate was found to be decreased in volunteers that followed intervention with prebiotics, such as wheat bran extract [69]. Fecal isovalerate and isobutyrate are produced due to the fermentation of valine, leucine and isoleucine and serve as precursors for fatty acid synthesis or as nitrogen donors for the production of other amino acids [70]. These two branched-chain fatty acids are often used as biomarkers of protein catabolism, but little is known about their impact on host health [71,72]. A limitation of the present study is that SCFAs were not measured in plasma, while the determination of markers of oxidative stress and inflammation could shed light on the underlying mechanisms connecting fatty acids profile with T1D.

The effects of pistachio described in the present study are applicable to a human diet. Certainly, if the amount of calories provided by pistachios are taken into consideration, 37.7 g (~38 g) of pistachios (i.e., 237.4 kcal or ~240 kcal) have to be included in a diet of 2000 kcal/per day. However, it has to be mentioned that the pattern of rat intestinal microbiota is different from that of humans. Specifically, at the phylum level, a higher

*Firmicutes-Bacteroidetes* ratio has been reported in humans. Human microbiota is mainly characterized by *Bacteroides* followed by *Ruminococcaceae* and *Clostridiales*, whereas in rats, the abundance of *Prevotella* is higher. In addition, fecal amounts of lactate are higher while those of acetate is lower in rats compared to humans [73]. In order to extrapolate the effects of dietary intervention or other factors from animal to human gut microbiota composition, such differences must be taken into consideration, and further studies must be conducted.

### 5. Conclusions

Dietary intervention with pistachio nuts for 4 weeks led to a beneficial alteration in plasma fatty acids composition since increased levels of MUFA were observed in STZ-induced diabetic rats. T1D resulted in decreased levels of 16:0, SFA and 18:3$\omega$3, along with a reduction in total $\omega$3 levels in adipose tissue; the ratio of $\omega$6/$\omega$3 was elevated in these animals as a result of the disease. Significant elevation of oleic acid, a dietary MUFA with health-promoting effects, and total MUFA in both diabetic and healthy rats were recorded after pistachio consumption for 4 weeks, while 18:2$\omega$6, PUFA and total $\omega$6 presented a decrease. On the other hand, fecal lactic acid was increased in healthy groups, indicating a relationship between an elevated presence of beneficial bacteria, as lactobacilli and lactate as their major metabolic end-product. Given the differences between the rat and human complex ecosystem that consists of the intestinal microbiota, the observed differences in an STZ- induced diabetic model are of great importance in order to carefully design human studies involving pistachio supplementation. According to the results of the present study, ~38 g of pistachios (or ~240 kcal) have to be included in a human diet of 2000 kcal/per day. Further studies need to be conducted to elucidate whether and how these results are associated with the actual clinical improvement and the entity of the disease, which is ultimately required to create better therapeutic approaches and management of T1D.

**Author Contributions:** Conceptualization, A.E.Y. and Y.K.; methodology, A.E.Y., N.K. (Nikolaos Kostomitsopoulos), N.K. (Nick Kalogeropoulos), V.T.K. and Y.K.; formal analysis, I.P. and A.E.Y.; investigation I.P., A.N. and A.E.Y.; resources, A.E.Y., N.K. (Nikolaos Kostomitsopoulos), N.K. (Nick Kalogeropoulos), V.T.K., Y.K. and E.B.; writing—original draft preparation, I.P. and A.E.Y.; writing—review and editing, I.P., A.E.Y., A.N., N.K. (Nikolaos Kostomitsopoulos), N.K. (Nick Kalogeropoulos), E.B., V.T.K. and Y.K.; supervision, A.E.Y. and Y.K.; project administration, A.E.Y. and Y.K.; funding acquisition, A.E.Y., V.T.K., Y.K. and E.B. All authors have read and agreed to the published version of the manuscript.

**Funding:** This research was funded by Greece and the European Union (European Social Fund—ESF) through the Operational Program "Human Resources Development, Education and Lifelong Learning 2014–2020" in the context of the project "Strengthening Human Resources Research Potential via Doctorate Research—2nd Cycle" (MIS 5000432). We acknowledge support of this work by the project "Research Infrastructure on Food Bioprocessing Development and Innovation Exploitation—Food Innovation RI" (MIS 5027222), which is implemented under the Action "Reinforcement of the Research and Innovation Infrastructure", funded by the Operational Program "Competitiveness, Entrepreneurship and Innovation" (NSRF 2014–2020) and co-financed by Greece and the European Union (European Regional Development Fund).

**Institutional Review Board Statement:** Not applicable.

**Informed Consent Statement:** Not applicable.

**Data Availability Statement:** Not applicable.

**Conflicts of Interest:** The authors declare no conflict of interest.

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
