# Peer review of "Dietary Pistachio (Pistacia vera L.) Beneficially Alters Fatty Acid Profiles in Streptozotocin-Induced Diabetic Rat"

_applsci, doi:10.3390/app12094606_

Round 1
Reviewer 1 Report
Manuscript ID: applsci-1695260 presents the study examining plasma fatty acid profiles and VAT as well as fecal SCFA after dietary intervention with pistachio nuts in streptozotocin-induced diabetic rats.
Line 110 reports “The amount of pistachio was added to yield a total of 100 g/kg fat in the pistachio diet”
The authors should specify what percentages of pistachios are added in the standard food, and how much food is provided to the animal daily. This data should be discussed later in the results and conclusions.
Tables should be checked for reported values and abbreviations in the text. The p-value must be written correctly.
Line 368 reports “fecal lactic acid was increased in healthy groups, indicating a relationship between elevated presence of beneficial bacteria as lactobacilli and lactate as their major metabolic end-product”. This effect of pistachio supplementation is possible on rats, but the composition of the rat intestinal microbiota is different from that of humans. The authors are invited to comment on this difference.
Are the effects of pistachio, described by the authors, applicable to the human diet? What amount should be consumed daily by humans?
This criticism should be addressed by the authors.
Author Response
Title: Dietary pistachio (Pistacia vera L.) beneficially alters fatty acid profiles in streptozotocin-induced diabetic rat (Manuscript ID: applsci-1695260)
Authors: Ioanna Prapa, Amalia E. Yanni, Anastasios Nikolaou, Nikolaos Kostomitsopoulos, Nick Kalogeropoulos, Eugenia Bezirtzoglou, Vaios T. Karathanos and Yiannis Kourkoutas
Response to reviewers’ comments
Authors would like to thank reviewers for their comments which advance the quality of the manuscript. Page and line numbers are referred to the revised version of the manuscript unless otherwise stated.
Reviewer 1
Manuscript ID: applsci-1695260 presents the study examining plasma fatty acid profiles and VAT as well as fecal SCFA after dietary intervention with pistachio nuts in streptozotocin-induced diabetic rats.
Line 110 reports “The amount of pistachio was added to yield a total of 100 g/kg fat in the pistachio diet”
The authors should specify what percentages of pistachios are added in the standard food, and how much food is provided to the animal daily. This data should be discussed later in the results and conclusions.
Authors agree with the reviewer’s comment. The percentage of pistachios in the food was 8% w/w and the daily amount of food which provided to the animals was 20g (standard food and pistachios).
On page 3, lines 117-118 the phrase: “The percentage of pistachios in the food was 8% w/w and the daily amount of food which provided to the animals was 20g (standard food and pistachios).” was added.
The phrase: “The amount of pistachios provided to the animals was based on the design of relevant studies including dietary interventions in animal models [30] and humans so that the energy content to be suitable for the human diet [31,32].” was added in the Materials and Methods section (Page 3, lines 118-121).
Τhe references: ”[30] Tsoukas, M. A.; Ko, B.-J.; Witte, T. R.; Dincer, F.; Hardman, W. E.; Mantzoros, C. S. Dietary Walnut Suppression of Colorectal Cancer in Mice: Mediation by MiRNA Patterns and Fatty Acid Incorporation. J. Nutr. Biochem. 2015, 26 (7), 776–783. https://doi.org/10.1016/j.jnutbio.2015.02.009.; [31] Li, Z.; Song, R.; Nguyen, C.; Zerlin, A.; Karp, H.; Naowamondhol, K.; Thames, G.; Gao, K.; Li, L.; Tseng, C.-H.; Henning, S. M.; Heber, D. Pistachio Nuts Reduce Triglycerides and Body Weight by Comparison to Refined Carbohydrate Snack in Obese Subjects on a 12-Week Weight Loss Program. J. Am. Coll. Nutr. 2010, 29 (3), 198–203. https://doi.org/10.1080/07315724.2010.10719834.; [32] Ukhanova, M.; Wang, X.; Baer, D. J.; Novotny, J. A.; Fredborg, M.; Mai, V. Effects of Almond and Pistachio Consumption on Gut Microbiota Composition in a Randomised Cross-over Human Feeding Study. Br. J. Nutr. 2014, 111 (12), 2146–2152. https://doi.org/10.1017/S0007114514000385” were added in the list of references.
In addition the phrase: “The effects of pistachio described in the present study are applicable to human diet. Certainly, if the amount of calories provided by pistachios are taken into consideration, 37.7 g (~ 38g) of pistachios (i.e. 237.4 kcal or ~240 kcal) have to be included in a diet of 2000 kcal/per day.” has been added in the Discussion section (pages 9-10, lines 389- 392). Please see also the response to the last comment.
Conclusion part was improved and more information was provided in summary regarding the conclusions from this work. Certainly, the conclusion part was re-written as: “Dietary intervention with pistachio nuts for 4 weeks led to a beneficial alteration in plasma fatty acids composition since increased levels of MUFA were observed in STZ- induced diabetic rats. T1D resulted in decreased levels of 16:0, SFAs and 18:3ω3, along with a reduction in total ω3 levels, in adipose tissue; the ratio of ω6/ω3 was elevated in these animals as a result of the disease. Significant elevation of oleic acid, a dietary MUFA with health promoting effects, and total MUFA in both diabetic and healthy rats were recorded after pistachio consumption for 4 weeks, while 18:2ω6, PUFA and total ω6 presented a decrease. On the other hand, fecal lactic acid was increased in healthy groups, indicating a relationship between elevated presence of beneficial bacteria as lactobacilli and lactate as their major metabolic end-product. Given the differences between the rat and human complex ecosystem that consists the intestinal microbiota, the observed differences in a STZ- induced diabetic model are of great importance in order to carefully design human studies involving pistachio supplementation. According to the results of the present study ~ 38g of pistachios (or ~240 kcal) have to be included in a human diet of 2000 kcal/per day. Further studies need to be conducted to elucidate whether and how these results are associated with the actual clinical improvement and the entity of the disease which is ultimately required to create better therapeutic approaches and management of T1D.”
Tables should be checked for reported values and abbreviations in the text. The p-value must be written correctly.
Authors agree with the reviewer’s comment. Tables have been checked for reported values and abbreviations in the text. The p-value has been written correctly. All corrections are presented as track changes.
Line 368 reports “fecal lactic acid was increased in healthy groups, indicating a relationship between elevated presence of beneficial bacteria as lactobacilli and lactate as their major metabolic end-product”. This effect of pistachio supplementation is possible on rats, but the composition of the rat intestinal microbiota is different from that of humans. The authors are invited to comment on this difference.
Authors thank reviewer for the fruitful comment. The paragraph: ”However, it has to be mentioned that the pattern of rat intestinal microbiota is different from that of humans. Specifically, at phylum level, higher Firmicutes-Bacteroidetes ratio has been reported in humans. Human microbiota is mainly characterized by Bacteroides followed by Ruminococcaceae and Clostridiales whereas in rats the abundance of Prevotella is stronger. In addition, fecal amounts of lactate are higher while those of acetate are lower in rats compared to humans [73]. In order to extrapolate the effects of a dietary intervention or other factors from animal to human gut microbiota composition such differences must be taken into consideration and further studies must be conducted” was added in the Discussion section (pages 9-10, lines 392-400).
The reference: “[73] Nagpal, R.; Wang, S.; Solberg Woods, L. C.; Seshie, O.; Chung, S. T.; Shively, C. A.; Register, T. C.; Craft, S.; McClain, D. A.; Yadav, H. Comparative Microbiome Signatures and Short-Chain Fatty Acids in Mouse, Rat, Non-Human Primate, and Human Feces. Front. Microbiol. 2018, 9, 2897. https://doi.org/10.3389/fmicb.2018.02897” was added in the list of references.
Are the effects of pistachio, described by the authors, applicable to the human diet? What amount should be consumed daily by humans?
This criticism should be addressed by the authors.
Authors would like to thank reviewer for the comment. If the amount of calories provided by pistachios is taken into consideration (i.e. 50.7 kcal in 427.1 kcal of rat diet), about 237.4 kcal/day originating from pistachio have to be included in a 2000 kcal of human diet. Pistachios which were used in the study provide 629.9 kcal/100g, so an amount of 37.7g has to be consumed daily.
The phrase: “The effects of pistachio described in the present study are applicable to human diet. Certainly, if the amount of calories provided by pistachios are taken into consideration, 37.7 g (~ 38g) of pistachios (i.e. 237.4 kcal or ~240 kcal) have to be included in a diet of 2000 kcal/per day.” has been added in the Discussion section (pages 9- 10, lines 389-392).
Reviewer 2
The problem undertaken at work entitled “Dietary pistachio (Pistacia vera L.) beneficially alters fatty acid profiles in streptozotocin-induced diabetic rat” is interesting, however, in the manuscript, there are some places that must be revised. Title clearly describes what the manuscript is about. However, abstract, methodology, statistical analysis and conclusion should be corrected. In addition, cited references not always correct.
Reviewer's suggestions below:
Regarding Abstract
The abstract should be expanded, information about research results, and work methodology should be added.
Authors agree with the reviewer’s comment. Abstract was expanded and work methodology was added.
Regarding Introduction
L.47-49:”Diet is a modifiable factor of pivotal importance which can significantly contribute in accomplishing better management of the disease.” – Please provide the reference.
Authors agree with the reviewer’s comment. The reference: ” [8] Hamdy et al., 2016” has been added on page 2, line 51 and in the list of references (Hamdy, O.; Barakatun-Nisak, M.-Y. Nutrition in Diabetes. Endocrinol. Metab. Clin. North Am. 2016, 45 (4), 799–817. https://doi.org/10.1016/j.ecl.2016.06.010.)
Introduction should be based on the information from other scientific papers (works), not on the authors' analysis.
Authors agree with the reviewer’s comment. Relevant references have been added in the Introduction section. Certainly the references: “[8] Hamdy et al., 2018; [22] Paun et al., 2016; [28] Garonzi et al.,2021 ” have been added on page 2, line 51, page 3 line 74 and line 92 and in the list of references (Hamdy, O.; Barakatun-Nisak, M.-Y. Nutrition in Diabetes. Endocrinol. Metab. Clin. North Am. 2016, 45 (4), 799–817. https://doi.org/10.1016/j.ecl.2016.06.010; Paun, A.; Danska, J. S. Modulation of Type 1 and Type 2 Diabetes Risk by the Intestinal Microbiome: Role of Gut Microbiome in Diabetes. Pediatr. Diabetes 2016, 17 (7), 469–477. https://doi.org/10.1111/pedi.12424.; Garonzi, C.; Forsander, G.; Maffeis, C. Impact of Fat Intake on Blood Glucose Control and Cardiovascular Risk Factors in Children and Adolescents with Type 1 Diabetes. Nutrients 2021, 13 (8), 2625).
L53-57. “Pistachios contain high amounts of monounsaturated fatty acids (MUFA) and especially oleic acid…, as well as polyunsaturated fatty acids (PUFA) [11].” – more detail is needed. How many MUFA, PUFA, oleic acids and SFA? Please add.
Authors agree with the reviewer’s comment. The text was re-written as: “Pistachios contain high amounts (73%) of monounsaturated fatty acids (MUFA) and especially oleic acid (70%), known for its cardioprotective, hypocholesterolemic, hypoglycemic and anti-inflammatory properties [10,11], as well as polyunsaturated fatty acids (PUFA) (16%) [12]. The presence of unsaturated fatty acids which is accompanied by low saturated fatty acids (SFA) content (11%), renders pistachio an optimal choice for maintaining a healthy lipid profile [13,14].”
L.68: “by our research group” – This information should be removed
The information was removed.
L.71-71: - – information on literature sources is missing
The information was added. The reference “[22] Paun et al., 2016” was added on page 3, line 74 and in the list of references (Paun, A.; Danska, J. S. Modulation of Type 1 and Type 2 Diabetes Risk by the Intestinal Microbiome: Role of Gut Microbiome in Diabetes. Pediatr. Diabetes 2016, 17 (7), 469–477. https://doi.org/10.1111/pedi.12424.)
L.87-90 – – information on literature sources is missing
The information was added. The reference “[28] Garonzi et al., 2021” was added on page 3, line 92 and in the list of references (Garonzi, C.; Forsander, G.; Maffeis, C. Impact of Fat Intake on Blood Glucose Control and Cardiovascular Risk Factors in Children and Adolescents with Type 1 Diabetes. Nutrients 2021, 13 (8), 2625).
Regarding Materials and Methods
L102-104: „The present study is a part of a larger study …. was examined [19].” – This information should be removed, because it does not apply to the submitted work.
The information was removed.
This part should be suplemented, more information about diet is needed. Why “control diet” was supplemented with corn oil (not another oils?)
Authors agree with the reviewer’s comment. More information about diet were added. The phrase: “The amount of pistachios provided to the animals was based on the design of relevant studies including dietary interventions in animal models [30] and humans so that the energy content to be suitable for the human diet [31,32].” was added in the Materials and Methods section (Page 3, lines 118-121).
Τhe references: ”[30] Tsoukas, M. A.; Ko, B.-J.; Witte, T. R.; Dincer, F.; Hardman, W. E.; Mantzoros, C. S. Dietary Walnut Suppression of Colorectal Cancer in Mice: Mediation by MiRNA Patterns and Fatty Acid Incorporation. J. Nutr. Biochem. 2015, 26 (7), 776–783. https://doi.org/10.1016/j.jnutbio.2015.02.009.; [31] Li, Z.; Song, R.; Nguyen, C.; Zerlin, A.; Karp, H.; Naowamondhol, K.; Thames, G.; Gao, K.; Li, L.; Tseng, C.-H.; Henning, S. M.; Heber, D. Pistachio Nuts Reduce Triglycerides and Body Weight by Comparison to Refined Carbohydrate Snack in Obese Subjects on a 12-Week Weight Loss Program. J. Am. Coll. Nutr. 2010, 29 (3), 198–203. https://doi.org/10.1080/07315724.2010.10719834.; [32] Ukhanova, M.; Wang, X.; Baer, D. J.; Novotny, J. A.; Fredborg, M.; Mai, V. Effects of Almond and Pistachio Consumption on Gut Microbiota Composition in a Randomised Cross-over Human Feeding Study. Br. J. Nutr. 2014, 111 (12), 2146–2152. https://doi.org/10.1017/S0007114514000385” were added in the list of references.
In addition the phrase: “The effects of pistachio described in the present study are applicable to human diet. Certainly, if the amount of calories provided by pistachios are taken into consideration, 37.7 g (~ 38g) of pistachios (i.e. 237.4 kcal or ~240 kcal) have to be included in a diet of 2000 kcal/per day.” has been added in the Discussion section (pages 9-10, lines 389-392). Please see also the response to the last comment.
Control diet was supplemented with corn oil because it has low oleic acid content which is the predominant fatty acid in pistachio nuts. In addition is the preferred oil used in such dietary interventions (Tsoukas et al., 2015). The phrase: “Corn oil was used in control diet because it has low oleic acid content which is the predominant fatty acid in pistachio nuts” was added in page 3, lines122-124.
Methods: in this part, the information about the used apparatuses should be added (e.q.water bath, centrifug, homogenizer), e.q. type, manufacturer, city, country
The information about the used apparatuses has been added (page 3, lines 141-142,145,148 and page 4 lines 152,153).
Hewlett-Packard 6890 Gas Chromatograph – should be added an information about the devices type, manufacturer, city, country
The information has been added (page 4, lines 152,153).
Regarding Results
L.195: „a: P<0.05 vs. DCD. N/D: Non-Detected.” - this paragraph is not clear and should be corrected.
The paragraph was corrected.
Table 1.- the used abbreviations should be explained under the table. Moreover, the results of the statistical analysis of the presented results are missing.
Authors agree with the reviewer’s comment. The used abbreviations were explained under the table. Authors kindly ask the reviewer to specify If the results of statistical analysis are not still clearly presented.
Table 2. statistics markings are not clear, please do correct. Similarly table 3. In addition, records such as: „1.345±0.625d”are not correct (1.345 d±0.625 – this record is correct)
Authors agree with the reviewer’s comment. Statistics markings and other records were corrected as indicated.
Regarding Conclusion
In the conclusion part, minor improve should be made. Not all information in summary does present conclusions from this work.
Authors agree with the reviewer’s comment. Conclusion part was improved and more information was provided in summary regarding the conclusions from this work. Certainly, the conclusion part was re-written as: “Dietary intervention with pistachio nuts for 4 weeks led to a beneficial alteration in plasma fatty acids composition since increased levels of MUFA were observed in STZ- induced diabetic rats. T1D resulted in decreased levels of 16:0, SFAs and 18:3ω3, along with a reduction in total ω3 levels, in adipose tissue; the ratio of ω6/ω3 was elevated in these animals as a result of the disease. Significant elevation of oleic acid, a dietary MUFA with health promoting effects, and total MUFA in both diabetic and healthy rats were recorded after pistachio consumption for 4 weeks, while 18:2ω6, PUFA and total ω6 presented a decrease. On the other hand, fecal lactic acid was increased in healthy groups, indicating a relationship between elevated presence of beneficial bacteria as lactobacilli and lactate as their major metabolic end-product. Given the differences between the rat and human complex ecosystem that consists the intestinal microbiota, the observed differences in a STZ- induced diabetic model are of great importance in order to carefully design human studies involving pistachio supplementation. According to the results of the present study ~ 38g of pistachios (or ~240 kcal) have to be included in a human diet of 2000 kcal/per day. Further studies need to be conducted to elucidate whether and how these results are associated with the actual clinical improvement and the entity of the disease which is ultimately required to create better therapeutic approaches and management of T1D.”
L.359-361: “According to our knowledge, this is the first …..STZ-induced T1D rat model” - this information should be removed - these are not the conclusions of the presented study.
The information was removed.
Reviewer 2 Report
The problem undertaken at work entitled “Dietary pistachio (Pistacia vera L.) beneficially alters fatty acid profiles in streptozotocin-induced diabetic rat” is interesting, however, in the manuscript, there are some places that must be revised. Title clearly describes what the manuscript is about. However, abstract, methodology, statistical analysis and conclusion should be corrected. In addition, cited references not always correct.
Reviewer's suggestions below:
Regarding Abstract
The abstract should be expanded, information about research results, and work methodology should be added.
Regarding Introduction
L.47-49:”Diet is a modifiable factor of pivotal importance which can significantly contribute in accomplishing better management of the disease.” – Please provide the reference.
Introduction should be based on the information from other scientific papers (works), not on the authors' analysis.
L53-57. “Pistachios contain high amounts of monounsaturated fatty acids (MUFA) and especially oleic acid…, as well as polyunsaturated fatty acids (PUFA) [11].” – more detail is needed. How many MUFA, PUFA, oleic acids and SFA? Please add.
L.68: “by our research group” – This information should be removed
L.71-71: - – information on literature sources is missing
L.87-90 – – information on literature sources is missing
Regarding Materials and Methods
L102-104: „The present study is a part of a larger study …. was examined [19].” – This information should be removed, because it does not apply to the submitted work.
This part should be suplemented, more information about diet is needed. Why “control diet” was supplemented with corn oil (not another oils?
Methods: in this part, the information about the used apparatuses should be added (e.q.water bath, centrifug, homogenizer), e.q. type, manufacturer, city, country
Hewlett-Packard 6890 Gas Chromatograph – should be added an information about the devices type, manufacturer, city, country
Regarding Results
L.195: „a: P<0.05 vs. DCD. N/D: Non-Detected.” - this paragraph is not clear and should be corrected.
Table 1.- the used abbreviations should be explained under the table. Moreover, the results of the statistical analysis of the presented results are missing.
Table 2. statistics markings are not clear, please do correct. Similarly table 3. In addition, records such as: „1.345±0.625d”are not correct (1.345 d±0.625 – this record is correct)
Regarding Conclusion
In the conclusion part, minor improve should be made. Not all information in summary does present conclusions from this work.
L.359-361: “According to our knowledge, this is the first …..STZ-induced T1D rat model” - this information should be removed - these are not the conclusions of the presented study.
Author Response

(The authors gave the same response as above.)

Round 2
Reviewer 1 Report
Authors have taken in account the comments and the new revised version of the manuscript has been improved.
Therefore, the revised manuscript meets the criteria required for pubblications.